# Implementation of an In-House 3D Manufacturing Unit in a Public Hospital’s Radiology Department

**DOI:** 10.3390/healthcare10091791

**Published:** 2022-09-16

**Authors:** Ruben I. García, Ines Jauregui, Cristina del Amo, Ainhoa Gandiaga, Olivia Rodriguez, Leyre Margallo, Roberto Voces, Nerea Martin, Inés Gallego, Rikardo Minguez, Harkaitz Eguiraun

**Affiliations:** 13D Printing and Bioprinting Laboratory, Biocruces Bizkaia Health Research Institute, Plaza Cruces s/n, 48903 Barakaldo, Spain; 2Department of Graphic Design and Engineering Projects, Faculty of Engineering in Bilbao, University of the Basque Country UPV/EHU, Plaza Ingeniero Torres Quevedo 1, 48013 Bilbao, Spain; 3Innovation and Quality Department, Cruces U. Hospital, Plaza Cruces s/n, 48903 Barakaldo, Spain; 4Radiology Department, Cruces U. Hospital, Plaza Cruces s/n, 48903 Barakaldo, Spain; 5Maxillofacial Department, Cruces U. Hospital, Plaza Cruces s/n, 48903 Barakaldo, Spain; 6Cardiovascular Department, Cruces U. Hospital, Plaza Cruces s/n, 48903 Barakaldo, Spain; 7Traumatology Department, Cruces U. Hospital, Plaza Cruces s/n, 48903 Barakaldo, Spain; 8Research Centre for Experimental Marine Biology & Biotechnology, University of the Basque Country PiE-UPV/EHU, Areatza Pasealekua 47, 48620 Plentzia, Spain

**Keywords:** public healthcare, radiology, sustainable development, additive manufacturing

## Abstract

Objective: Three-dimensional printing has become a leading manufacturing technique in healthcare in recent years. Doubts in published studies regarding the methodological rigor and cost-effectiveness and stricter regulations have stopped the transfer of this technology in many healthcare organizations. The aim of this study was the evaluation and implementation of a 3D printing technology service in a radiology department. Methods: This work describes a methodology to implement a 3D printing service in a radiology department of a Spanish public hospital, considering leadership, training, workflow, clinical integration, quality processes and usability. Results: The results correspond to a 6-year period, during which we performed up to 352 cases, requested by 85 different clinicians. The training, quality control and processes required for the scaled implementation of an in-house 3D printing service are also reported. Conclusions: Despite the maturity of the technology and its impact on the clinic, it is necessary to establish new workflows to correctly implement them into the strategy of the health organization, adjusting it to the needs of clinicians and to their specific resources. Significance: This work allows hospitals to bridge the gap between research and 3D printing, setting up its transfer to clinical practice and using implementation methodology for decision support.

## 1. Introduction

Health 4.0 has emerged as a new concept based on the development of new enabling technologies, such as: medical Cyber-Physical Systems (medical CPS), big data analytics, machine learning, blockchain, virtual reality and 3D printing. The results of implementing these new technologies have been reported regarding benefits that improve the quality, flexibility, productivity, cost-effectiveness or reliability of healthcare services, in addition to increasing patients’ satisfaction. However, developing and applying Health 4.0 technologies is a non-trivial and complex endeavor [1]. Additive Manufacturing (AM) or three-dimensional (3D) printing is one of the newest technologies introduced into the healthcare system, and the concepts considered to be related to this area are described below.

### 1.1. Clinical Utility and Indication

A thorough evaluation and understanding of the relationship between tumors, organs and adjacent vasculature are essential for operative planning. Advances in medical imaging have provided more accurate diagnostic tools by adding a third dimension to conventional techniques. However, these conventional techniques only show two-dimensional images, relegating interpretation mainly to the surgeon’s visuo-spatial ability. Through AM, we overcome the interpretation dependency barrier by introducing, in addition, the sense of touch [1].

Performing a search in the Pubmed portal with keywords associated with “Additive Manufacturing”, we found an increasing evolution of the number of publications in recent years (Figure 1).

After our search and filtering of meta-analyses and systematic reviews from the last 5 years, we found up to 210 results. Most of these results support the value of 3D printing in up to 9 specialties and more than 30 different pathologies. Summarizing, they consider that AM in medicine can provide clinical utility as a support tool for complex surgical procedures [2], improving the surgical treatment and reducing the number of derived complications, the surgical time, the volume of blood lost during surgery and the number of intraoperative X-rays. These clinical benefits are not directly associated with clinical specialties but depend largely on the target pathology [3,4,5].

The clinical variability linked to factors such as training, experience [6] or surgical equipment available for the clinical team has led most studies to justify medical AM from a qualitative rather than quantitative point of view [7,8]. As a consequence, some authors [9] have questioned the scientific rigor and the lack of conclusive clinical data of these applications. Although greater scientific diligence is observed in fields such as oral and maxillofacial surgery and the musculoskeletal system [10], there are still questions regarding the need for more methodologically rigorous evaluations for many general applications [11], more randomized and controlled Level 1 clinical trials and studies with longer follow-up periods and larger sample sizes [10].

Contrary to some of the doubts mentioned, the advantages observed by clinical teams have boosted the use of AM in clinical practice [12]. In addition, the great scientific-technological advances of the last decades have led to technological democratization and, consequently, to the emergence of in-house 3D printing laboratories in hospitals. These laboratories, also known as point-of-care manufacturing laboratories (POCMLs), allow hospitals to be manufacturers of printed customized anatomical models, surgical guides, personalized prostheses and other applications [13,14].

Considering all these aspects, the Radiological Society of North America (RSNA) created a “Special Interest in 3D Printing” team to elaborate guidelines for medical 3D printing. These guidelines describe recommendations for the consistent and safe production of 3D models derived from medical images [12]. 

### 1.2. Economic Issues

Cost–benefit justification is mandatory in order to adopt any kind of technology in the hospital environment. In some economic studies, medical 3D printing decreases operating room costs due to reduced surgical time [4]. The AM process in medicine includes several steps [15], but unfortunately, relevant details such as the manufacturing cost of these models or indirect costs such as technician times are not well reported [16,17]. This fact hinders the decision regarding technology adoption in hospitals [18]. The great variability in terms of the processes used (outsourced or in-house manufacturing) [19] also has a major impact on the economic aspect [20].

Since the implementation of AM is a strategic decision to be made by hospital managers [20], it is necessary to identify and locally define the critical success factors (CSFs): environment analysis, resource analysis and strategy evaluation [21]. According to Chaudhuri et al. [22], hospitals need to consider other aspects, such as the complexity of the surgical planning processes, critical delivery time, annual demand and prioritization of hospital objectives.

### 1.3. “In-House”/Point-of-Care Manufacturing

Some authors suggest that point-of-care (POC) 3D printing offers a cost-effective and less time-consuming process [23,24]. There are other factors that led hospital management to create POCMLs, for instance, technological democratization, the need for agility in communication and speedy response in urgent cases, dependence on in-house clinical knowledge and security gaps when sharing clinical information [13].

These needs coexisted in a non-specific legal framework until there were legislation updates (different for each country) [25]. Updates of regulations governing the manufacture of customized medical devices are becoming increasingly stringent [26]. In Europe, Medical Device Regulation 2017/745 (MDR) treats AM devices and accessories as “custom-made-devices”. These devices must be prescribed by the clinician, under their responsibility, and intended for the sole use of a particular patient. Additionally, to implement a POCML, clinicians need to develop their own in-house regulatory quality management system protocols to control the processes.

### 1.4. Leading Digital Transformation

Due to the flexibility and increasing availability of low-cost AM, it is expected that its applications for medical radiology will continue increasing [27], with more users purchasing this type of equipment as office supplies. 

In any case, the human factor in these digital transformation processes [28] and the technological adoption resistance that may be present in the most conservative clinicians [10] must be taken into account when implementing a POCML. Considering the hospital complexity, clear leadership and commitment from managers are necessary [29,30]. In the Implementation Science context, these leaders are also known as “champions” [31].

Beyond the leadership profile, to maintain patient safety, it is important that the personnel involved in this process are proficient in medical imaging and 3D printing and have knowledge of quality control programs [12].

According to different authors [32,33], the use of standards is mandatory to develop an orderly and evaluated implementation, analyzing the impact and long-term economic viability according to the particular characteristics of each organization [22].

The aim of this study was the evaluation and implementation of a 3D printing technology service in a radiology department. The results and resources were measured and adjusted accordingly, attending to the particular needs of our hospital. The main scientific contribution of the present work is that after a period of 6 years, we developed a methodology and successfully implemented it in a complex system of a public hospital, which covers the gap between research and clinical practice, allowing the technology transfer of AM technology for the patient’s benefit.

## 2. Materials and Methods

This work was performed at the Cruces University Hospital (CUH), which belongs to an Integrated Services Organization named OSI Ezkerraldea Enkarterri Cruces that includes 11 Primary Care Units (Osakidetza—Basque Health System). This public tertiary-level hospital provides health assistance as a reference center inside and outside the Basque Country (Spain). The hospital has 847 beds and 33 operating theaters and performs more than 31.000 surgical operations per year.

This bottom-up project was approved by the CUH Directors in a 2016 Innovation Committee and had a direct follow-up during its execution.

We consider important to clarify that the present study is not intended to be a comparative clinical study, but rather a technological implementation study. We defined a workflow led by the radiology department of our hospital based on the existing literature and with different implementation strategies [31,34]. We analyzed and evaluated the impact of the technology during a period of six years and progressively scaled up the resources based on activity fluctuations. The most relevant packages are described below.

### 2.1. Workflow Definition

The workflow defined for our hospital is described in Figure 2. There are different steps and personnel involved in the medical 3D printing process and the manufacturing of anatomical model processes or customized devices for each patient.

The activity was integrated into the workflow of the radiology department, allowing expert clinicians in the area to perform an initial screening based on the appropriateness of the clinical indication for each request.

### 2.2. Human Resources

For technology implementation, leaders must be selected. The chosen leader was a biomedical engineer working in the innovation and quality department of the aforementioned hospital and had an academic background in medical 3D printing.

Another clinical leader was defined due to the importance of the radiology department in the 3D printing process: a senior radiologist with a permanent position and experience in this hospital was designated. In addition, other junior radiologists in their 3–4th year as Resident Medical Interns (RMI) were included in the team. This activity was included in their academic objectives. Through agreements with professional training schools, an academic program was also developed to prepare radiodiagnostic technicians in 3D printing techniques, incorporating these new profiles into the team. In order to obtain the certificate and course credits, clinicians had to complete a satisfaction survey.

### 2.3. Segmentation Software

Medical image segmentation should be performed through software that is validated and integrated into clinical practice. We used Philips Intellispace Portal V11.1 (3D modeling application) and GE AW Server 3.2, both with direct access to the picture archiving and communication system (PACS). 

### 2.4. Online Application System

A request website on the hospital intranet was developed to receive, manage and monitor the activity. Clinicians have to fill in an application form with the following questions regarding the 3D model: contact information, patient code or reference, sterilization requirements, type of surgery and usage. The 3D radiologists and engineers receive an e-mail alert, and the petition website allows the tracking of the status of the request (segmented or not).

### 2.5. Additive Manufacturing

During the project, attending to specific demands, different 3D printers involving different technologies were acquired: (i) Fused Deposition Modeling (FDM) technology: Ultimaker2+ in 2016 (Ultimaker, Ultrecht, Netherlands). (ii) Stereolithography (SLA): Form 2 in 2018 (Formlabs, Somerville, MA, USA). (iii) Material Jetting (MJ): Objet 260 Connex3 in 2019 (Stratasys, Eden Prairie, MN, USA). Each printer was used with its commercial software.

### 2.6. Quality Processes

We defined a standard operating procedure (SOP) for the quality control process of every manufactured anatomical model, regardless of the technology or material.

The quality of each performed activity was evaluated by two fundamental aspects:
(i)The speed of response of the service: To analyze urgency and responsiveness, the request date, the delivery requirement date and the final delivery date were recorded.(ii)The technical quality of the products manufactured, or errors made in 3D printing: For the manufactured quality, we measured the absolute error between the physical model and the unprocessed virtual model or STL file. For each manufactured model, we took 6 dimensions (3 for each Cartesian coordinate) in both the physical model (with a digital Vernier caliper) and the virtual model (with the ^®^Meshmixer software; version 3.5.474, 2017 Autodesk, Inc., San Rafael, CA, USA). 

All non-conformities must be recorded to comply with FDA Guidelines [35], considering these the most relevant ones: image error, manufacturing error, dimensional error or consideration of device use. Additionally, all information about the application, segmentation, manufacturing and quality requirements was recorded in a database.

### 2.7. Satisfaction and Usability Surveys

In the last year of this study (2021), for each application, we sent a usability and satisfaction survey to be completed by the clinician. This was a voluntary survey to specifically assess each case and to obtain the needed feedback for further service improvements. The questions from this survey are described in Appendix A and were prepared based on similar experiences found in the scientific literature [36,37]. Questions were answered on a Likert scale: 1 (Strongly disagree)–5 (Strongly agree).

The purpose of this survey was to obtain the qualitative perception of the service and its use once the clinician utilized the device in the operating theater. Therefore, the survey included generic questions that may be used for any type of pathology and department.

### 2.8. Costs

To estimate the costs of the process, direct and indirect costs were considered. The direct costs were calculated by computing the material expenses and the working time of the different technicians involved in each case. The cost of the human resources was provided by the Economic Evaluation Department of our hospital. In the case of the amortization of the equipment, considered indirect costs, it was computed by taking into account the capital cost divided by 1592 h of annual machine work time during a 6-year period of amortization.

In order to obtain cost–benefit conclusions, other clinical parameters must be considered. These parameters had to demonstrate that the technology generates better patient outcomes. To obtain this type of result, prospective clinical trials with strict inclusion criteria (by pathology) that allow a pairwise comparison are required.

## 3. Examples of Applications

The following are different examples of activities carried out by the 3D manufacturing unit created.

According to the workflow defined above, an anatomical model was generated based on a DICOM medical CT image, and segmentation was performed by an expert radiologist. The printing technology selected was dependent on the request and, in all cases, managed by the engineering team. 


a.
*Anatomical model for planning, training, teaching and patient communication of obstructive hypertrophic cardiomyopathy (cardiovascular surgery)*



Obstructive hypertrophic cardiomyopathy (OHC or HOCM) is a pathology where the cardiac walls thicken and hinder blood pumping. This surgical treatment requires open surgery for resection of the excessive tissue [38]. The resected pieces should measure around 5 to 10 mm. There are technical limitations due to poor vision and risks of serious complications, which means that, sometimes, the desired objective might not be achieved (Figure 3A). Scarce resection means a surgical failure, but excessive resection can induce a ventricular connection, leading to a fatal result.

The anatomic variability of the heart is vast, and there exist many structures that are very close, which can be easily identified with a radiology image but not during the surgical activity. Thus, these OHC surgeries are complex procedures that depend largely on the skill and expertise of the surgical team.

Segmentation is performed according to the following anatomical structures: 1 long axis (2 pieces) and 2 short axes/perpendicular cutting planes (3 pieces: basal, middle and apex) (Figure 3B,C). A 3D-printed heart anatomical model is used as a support tool for planning and as a reference during the intervention. Myocardial resection is performed using a marked diathermy loop based on measurements taken on the 3D model prior to the surgery (Figure 3D).

Such planning and personalization of the treatment reduce complications, as well as clinical variability and surgeon dependence. The surgical technique based on 3D models provides safety and confidence to the clinician, allowing a more aggressive and safe surgery, as well as better clinical results. The 3D model also allows better communication with the patient, achieving greater tranquility on their part, seeing that the surgery is being planned in a personalized way [38]. 


b.
*Anatomical model and cutting guides for surgical treatment of a complex benign neoplasm (maxillofacial surgery)*



Although there are many complex neoplasms that can grow in the jaws (odontogenic tumors, such as ameloblastomas, and osteolytic lesions, such as metastatic tumors), the most frequent are odontogenic cysts (OCs). An OC is a pathological, epithelial-lined cavity containing fluid or semi-fluid that arises from the epithelial remnants of tooth formation. Odontogenic cysts are often asymptomatic and therefore may grow before any clinical signs are noted, so their presence is often an incidental finding on radiographic examination. Most OCs are treated by either enucleation or marsupialization. Enucleation involves the complete removal of the cyst and is the treatment of choice [39]. This technique involves the creation of a surgical window in the wall of the cyst, decompression of the cyst and removal of its content. In both cases, it is mandatory to access the cyst in the most direct way, avoiding injury to soft tissues or other anatomical structures, such as nerves or vessels, to perform an adequate treatment. Access to compromised anatomical sites, such as the jaw’s bony interior, poses a permanent challenge in modern medicine despite advances in diagnostic and surgical techniques. (Figure 4).

Three-dimensional technology reduces clinical variability and improves precision in this kind of surgical procedure. Through surgical tools designed for specific patients, it is possible to obtain better precision and intra-surgical references [40]. These tools, known as Patient-Specific Instrumentation (PSI) or cutting guides, are used in different surgical procedures in different specialties.


c.
*Anatomical model for pre-modeling synthesis plates for surgical treatment of acetabular fractures (traumatological surgery)*



The treatment of pelvic fractures involves a complicated approach for the osseous pelvis, which contains viscera and large vessels that make handling difficult. These fractures require complex bone reconstruction using osteosynthesis plates, which must be perfectly adjusted to the patient’s anatomy. Current surgical methods entail a great loss of time during the operation, sometimes with a difficult and/or dangerous approach, which limits the final retouching of the plates. This final adjustment of the plates is performed at the surgeon’s discretion and requires great expertise [41]. The trial-and-error method necessary to obtain an appropriate molding presents the disadvantage of excessive operating time, which is not convenient for the patient. The combination of reverse engineering with AM can provide a solution to this problem. Thus, the plates can be molded before the operation based on a previously printed biomodel obtained directly from a scan of the patient, reducing the operating time and costs and providing benefits for the patient, such as less exposure and lower infection risk [41,42] (Figure 5).

The planning and personalization of the treatment result in a reduction in complications, as well as reductions in clinical variability and surgeon dependence. The surgical technique based on 3D models provides safety and confidence to the clinician and better clinical results, such as shorter operation time, less intraoperative blood loss and less intraoperative fluoroscopy. The 3D model is used, in addition to planning the surgery, to improve the coordination of the surgical team, to teach residents and to explain the process to the patient, which, as already mentioned in the previous cases, gives him/her peace of mind and confidence.

## 4. Results

### 4.1. Training Courses

Through the hospital’s Training Department, the biomedical engineer leader and the referring radiologist instructors imparted five “Hands-On Training 3D Printing Courses” (per year), certified by the “Basque Council of Health Professions”. A total of 32 (52%) of the 62 participants were from the Radiodiagnostic Department. The courses had an excellent impact, with an average overall course evaluation of 9.29 (maximum 10).

### 4.2. Evolution of Requests

We performed 352 cases in 6 years for 27 different hospital departments. Despite the COVID-19 outbreak in 2020 and 2021, during which annual surgeries in our hospital decreased by 25%, there was a consecutive annual request increase (Figure 6). 

A total of 83.2% of the requests were for a 3D-printed anatomical model, 10.2% required a device (prototype), and 6.3% required a patient-specific surgical guide, while only 0.3% required a patient-specific implant. The most requested anatomy was the heart (17%), followed by the mandible (16%) and the hemipelvis (10%) (Figure 7A). The requests were made by 85 different clinicians. As shown in (Figure 7B), 31% of the requests were performed for maxillofacial surgery, 21% were for cardiovascular surgery and 18% were for traumatology (Figure 7B).

Table 1 shows the different types of purposes of the 3D printing models. The same anatomical model may be used for multiple purposes, and each type of usage was quantified. More than half of the requests were made for surgical planning (54%), which is closely related to personalized surgery simulation (36%). Additionally, Table 1 shows the total number of uses, as well as the percentage of each type of use compared to the 352 total number of requests.

### 4.3. Quality Control

*(i) Response time*. One of the main aspects that justifies in-house manufacturing is the urgency response time. To analyze the urgency of the requests received, four levels of urgency were differentiated: Very Urgent (delivery within 3 days), Urgent (delivery within 3–7 days), Semi-urgent (delivery within 7–14 days) and Non-urgent (delivery after 14 days or delivery date not specified) (Figure 8).

During 2020 and 2021, Very Urgent and Urgent requests increased compared to previous years, meaning that more than 1/3 of the cases had to be delivered in less than 7 days. It would be impossible to satisfy this request demand without an in-house service (Figure 8).

The majority of cases (69.03%) were delivered on time, whereas in 17.9% of cases, the request was not urgent or did not require 3D printing (Figure 8). In addition, 1.14% of the requests were not performed due to technical problems, and the remaining 11.93% did not arrive on time. As shown in (Figure 9), since 2020, there has been a lower number of cases not completed or not delivered on time.

*(ii) Printing errors*. We did not detect non-conformities in the area of interest of the printed anatomy regarding printing errors. Methodological inspections were performed in a total of 196 cases. Table 2 shows the number of cases performed with each of the technologies as of 2019, with the service established in terms of staff and equipment, as well as the learning curve overcome. Only cases in which one technology was used exclusively (not combined) were analyzed. For each common technology (FDM, SLA and MJP), the measurements made on the model and STL and the average deviation (% error) that exists between the STL measurement and the measurement on the 3D-printed model are shown. The dimensional errors produced in the manufacture of the biomodels were under 3%.

### 4.4. Cost

The costs derived from the manufacture of different anatomies depend on the anatomical complexity and its dimensions. Table 3 shows the printing time and manufacturing costs of the three most requested anatomies. We selected three anatomies of medium-sized patients and simulated the material costs and printing time for each of these three anatomies using the three common technologies.

Table 4 shows both the costs and average time spent in each of the phases of the process: segmentation by a radiologist, design by an engineer and manufacturing by a manufacturing technician; the costs of the personnel linked to process operator times have a higher variability depending on the complexity of the anatomy. For example, the time used to segment a bone is the same whether it is a mandible or a pelvis. This is the reason why the table is grouped according to complexity rather than anatomy.

### 4.5. Questionnaires

Table 5 shows the results of the surveys completed voluntarily by 29 different solicitants during 2021. Twenty-four of them were senior clinicians, four were section chiefs and one was a resident. The results were very positive in terms of utility for planning (*n* = 25) in relation to the information available compared to digital models (4.36 ± 0.95). In addition, a positive outcome was experienced during planning using the model (4.32 ± 0.95). Similar results were obtained for questions related to communication with the patient (*n* = 13). Specifically, the model helped the patients or guardians to be more receptive and collaborative (4.15 ± 1.35) as well as to better understand the intervention (4.08 ± 1.31). The same outstanding score was observed for its usefulness during the surgery (*n* = 15), either because it improved communication between colleagues (4.73 ± 1.26), increased the confidence of the team (4.33 ± 1.34) or had been used during the surgery as a guidance tool (4.60 ± 1.25). The results also indicate that models represented the anatomical area as observed during the operation (4.39 ± 0.99) and that scaled models (1.43 ± 0.88) or digital files were not useful (1.68 ± 1.22).

## 5. Discussion and Conclusions

CUH has been working on implementing 3D printing technology since 2016.

We agree with other authors that it is important to define CSFs by analyzing the institution’s environment, resources and strategy [21,22]. This study can help further implementation strategies, not only in surgery but also in other areas related to 3D printing in healthcare, where challenges involving this transfer have already been reported [43]. We consider it essential to develop an implementation project that will dimension the needs of each hospital in a concrete manner following these steps.

### 5.1. Clinical Indication

According to the scientific literature, 3D printing is already in a phase of clinical transfer, becoming an enabling tool that evolves from medical imaging itself [19]. Although there are certain doubts about the methodological rigor of some publications, our experience agrees that the technology can provide a differential value, especially in complex cases [2,6,24]. Thus, it should be borne in mind that it is in these complex cases where finding a comparative pair within a higher-level clinical study may not be feasible in the short term. Even the bias that this type of study may have must be considered, considering that 3D printing represents a small part of the clinical process, in addition to the continuous innovation of surgical techniques [44]. In these cases, clinical experience or the evolution of the surgical treatments themselves may have more impact on the final patient outcome than any other variable introduced in the process. In any case, we consider it necessary to continue to generate high-level scientific evidence, especially in less frequent pathologies, but without losing focus on the value of any diagnostic support tool in a complex process.

For all these reasons, we believe that high-level hospitals should consider 3D printing as another support tool and work on the adequacy of the indication, taking into account the economic repercussions of the incorporation of new technologies.

### 5.2. In-House/Point-of-Care Manufacturing Decision

The decision of whether to create a POCML, to outsource the entire activity or to create a mixed process depends on the complexity of the surgical planning processes, the critical lead time, the annual demand and the hospital’s objectives [22]. Measuring the evolution of requests is key for decision making [30], but beyond that, we consider it essential to have a dashboard with more indicators, such as petitioners, clinical services, clinical applications, delivery dates and types of use. With this information and the usability forms, the urgency of the requests, the complexity and the expansion of the service can be better measured.

Regarding the expansion, it is noteworthy that our results show 352 requests made by 85 different clinicians from 27 different services, which shows that this is not a fad of a few early adopters [8]. The growth during the COVID-19 outbreaks also reflects the degree of implementation and routine use, a fact that may also have been motivated by the ease of requesting systems discussed below. The dynamization activity through training courses may also have helped to extend its use.

If the decision is made in favor of a POCML, it is mandatory to define the workflow and response capacity [12,45], as well as a product portfolio based on experience. Adapting a workflow to the resources of each organization can be the key factor in the success of the implementation [46,47]. Quantifying the error that can be assumed in the process [35,48] is important for clinical security, so it is essential to define all these “artisanal processes” in SOPs for each type of application, following the recommendations of local regulations (in our case, UNE-EN-ISO13485). In addition to this technical validation, we consider it important to perform a joint validation (clinician, radiologist and engineer), especially for models containing masses or soft tissues. Despite the fact that customized anatomical models or surgical guides may be part of the imaging evolution itself, the activity of manufacturing these customized medical devices must be performed according to the regulations of the country [26], such as MDR in Europe.

### 5.3. Leading Digital Transformation

We agree with the authors who proposed that radiologists have to be the center on which the implementation of AM technology in hospitals pivots [30,49]. Radiologists must be the leaders because of the precision required in the segmentation and the relevance of image acquisition processes in the final product. The interpretation of the image required in complex cases demands the competencies and clinical responsibility of the staff who are making this diagnostic decision [12]. In relation to the software and management of medical images, radiology services also have a fundamental role in following a two-way approach: assessing the indication of the request according to the service portfolio [12] and integrating the process into hospital information systems or picture archiving and communication systems (PACs) to ensure the control of medical information and personal patient data [23]. Medical images should be encrypted prior to outsourcing. Facilitating this type of process through user-friendly integrated software could prevent security breaches or technology disuse [50]. Therefore, we consider it important to develop a request manager within the Electronic Health Records (EHR) hosted in the radiology activity census for management and monitoring of the activity in an integrated and secure manner.

New profiles, such as biomedical engineers or radiology technicians trained in AM, can help during the implementation of AM technology. These new profiles may also be important for obtaining a manufacturing license and should have access to medical records as clinical staff. Creating a POCML entails a commitment to recruiting profiles, considering the responsibility in a sector as complex and sensitive as healthcare [51].

Human and technical resources should be adjusted to the needs of each organization [15,52] based on pilot results and the evolution of the activity. In any case, in order to have control based on the hospital’s investment, we recommend the creation of expert clinical committees [12] in the hospital to ensure the appropriateness of the clinical indication for use. In our case, the role of radiologists was the key to filtering dubious cases regarding the appropriateness of the clinical indication, leading to sustainable personalized medicine.

In the same way that the RSNA [12] created the special interest group on 3D printing for correct technology transfer, we consider it important to collaborate in national networks, with the aim of generating national guidelines and activity codes within radiology, through Scientific Societies. In addition, the establishment of indicators that allow for the assessment of the global process and for the integration of all professionals involved has been suggested by some studies [53]. However, the heterogeneity of the technology and of the surgical techniques and equipment themselves must be considered.

### 5.4. Technology Investment and Cost-Effectiveness

Manufacturing methods and processes are not standardized, so the costs derived from each process can be very different [24]. In addition, the rapid evolution of AM technologies implies that equipment will become obsolete within a few years. In-house knowledge allows choosing the most appropriate technology, avoiding market dependency and defining the optimal processes [54,55]. Buying mature, validated and stable manufacturing technologies is mandatory [56].

Manufacturing costs are highly dependent on the selected technologies and materials [19]. Therefore, fluent communication with the clinician is mandatory to understand the application and select the appropriate resources [55].

While these direct costs will have an economic impact on the surgery, indirect costs must also be considered, as concluded by other authors [16,17]. Segmentation performed by a specialized operator with integrated and validated software could reduce the cost of the AM process by saving time [57]. The rapid evolution of technologies, materials and software requires an annual training program oriented to clinicians, which will allow them to better understand the process, select the optimal technology and promote its use, streamlining the process and allowing cost-effective integration.

We agree with other authors who concluded that the procedure room time saved can potentially offset cost [58]. Although more specific studies are required, it is mandatory to adjust and measure investments locally. Considering that we studied complex cases, both in terms of the number of annual requests and the time invested per radiologist for each type of anatomy, we can conclude that: (i) in the early stages of implementation, the time invested by radiologists may be affordable with their own staff, and (ii) direct costs (materials) and in-direct costs (human resource time and machine amortization) that a 3D-printed anatomical model manufactured with a certified technology such as SLA may range between EUR 13.97 and EUR 81.14. We consider this cost acceptable for complex processes, where the cost of surgery may exceed EUR 10,000 (less than 1% of the surgical cost in the worst case). Perhaps more expensive technologies such as MJ may generate greater discussion regarding the cost added. Additionally, managers need to consider that technology offers person-centeredness attention [59] that humanizes healthcare, which goes beyond economic outcomes. In addition, more personalized training programs need to be developed in accordance with the conditions of each healthcare organization [60].

Therefore, the role played by radiologists is also key in the economic aspect, not only due to the value they provide in the image segmentation process but also for the identification and filtering of cases in which there are doubts about the appropriateness of the clinical indication for each case.

Although the results of this article are focused on anatomical models and surgical guides, the POCML has enabled the self-supply and manufacture of devices in situations of supply shortage, which has become increasingly important with the current supply crisis [61].

The literature shows that AM has become a tool that provides high value in multiple surgical processes as an innovative enabling tool in medical imaging. The heterogeneity of procedures, software, technologies and materials has led to doubts as to the methodological value of scientific studies that justify its implementation in hospitals, both at the clinical level and in terms of economic evaluation. The catalog of manufacturing equipment is growing more user-friendly and economical, which has led to its democratization and implementation without standards, which generates clinical security gaps [62]. In any case, the technology is already part of the standard surgical tools used by many professionals, and it is therefore important to manage its use in hospitals.

Thus, it is imperative that healthcare organizations develop a 3D technology implementation strategy, and considering having an in-house manufacturing laboratory is appropriate. Thus, it is necessary to evaluate the impact at the local level depending on the activity and order this use within a legal framework for the in-house manufacture and/or outsourcing of customized medical devices. This implementation must be accompanied by education and training of healthcare professionals, with special attention to radiology departments, which are the key service to lead this implementation in an orderly and cost-effective manner.

## Figures and Tables

**Figure 1 healthcare-10-01791-f001:**
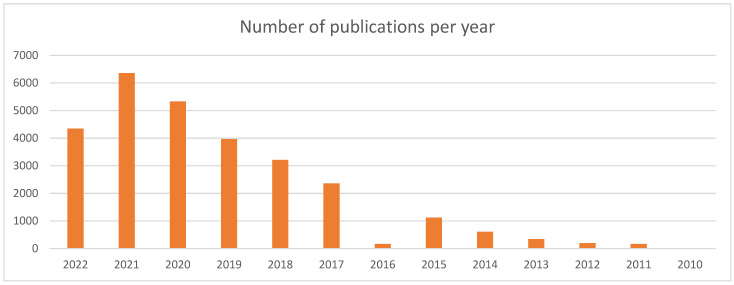
Search of Pubmed for articles related to medical 3D printing in recent years. Search query: “Three-dimensional printing” OR “3D-Printed” OR “3d-print” OR “Threeprint” OR “3D print” OR “rapid prototyping” OR “additive manufacturing”.

**Figure 2 healthcare-10-01791-f002:**
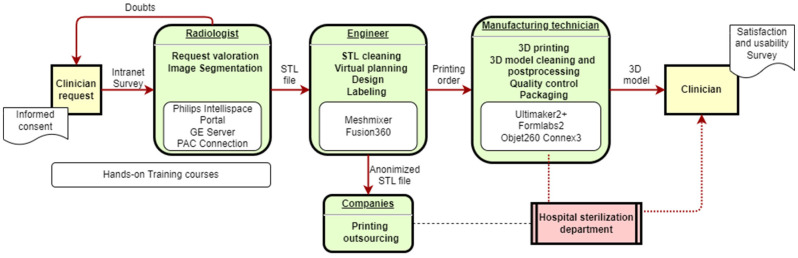
Workflow defined for the 3D printing process at Cruces Hospital. The request is made by the clinician through a form on the intranet. The request is received by the 3D printing team formed by radiologists and engineers, the request is evaluated, and once the CT or MRI image test has been performed, the segmented files are sent in STL format to the engineers who are responsible for designing and 3D image processing to ensure its correct printability. The printing, processing, packaging and quality control of the model is performed by the manufacturing technician. In the case of operating room use, the model is sent to the sterilization service.

**Figure 3 healthcare-10-01791-f003:**
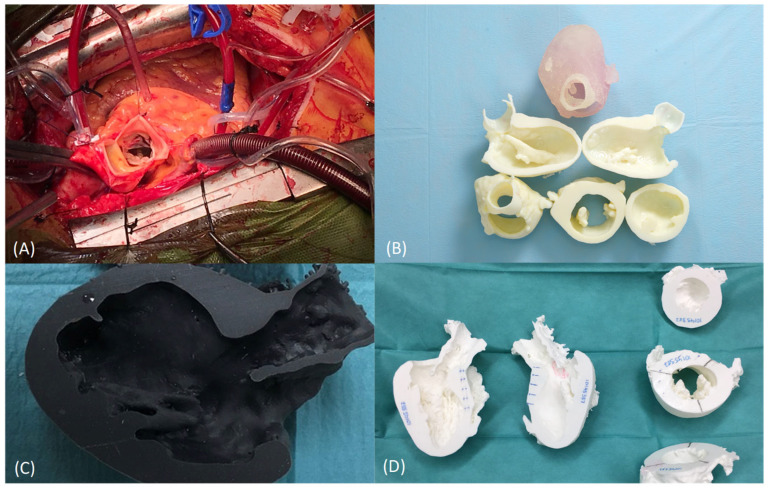
Anatomical model for planning, training, teaching and patient communication of obstructive hypertrophic cardiomyopathy (cardiovascular Surgery). (**A**) Surgical field and intraoperative vision in HCM surgical treatment. (**B**) Three-dimensional anatomical models for HCM planning, with pieces that represent the short axis and 3 chambers, manufactured by MJ technology. (**C**) A 3D anatomical model for HCM planning, showing a piece that represents longitudinal cutting of one chamber, manufactured by SLA technology. (**D**) Height and depth references of the resection to be performed on the cardiac wall, painted on the printed anatomical biomodel. References are painted on the diathermy loop itself.

**Figure 4 healthcare-10-01791-f004:**
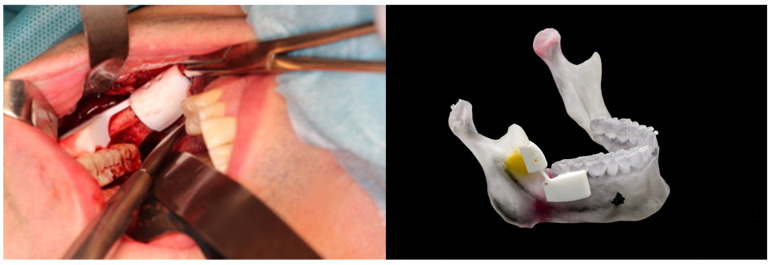
Anatomical model and cutting guides for surgical treatment of a complex benign neoplasm (Maxillofacial Surgery). Left: Surgical field and intraoperative vision with the surgical guide in place. Right: Surgical guide in polyamide material manufactured with SLS technology. Anatomical model of the jaw and neoplasms manufactured by MJ technology for planning the surgery based on DICOM images.

**Figure 5 healthcare-10-01791-f005:**
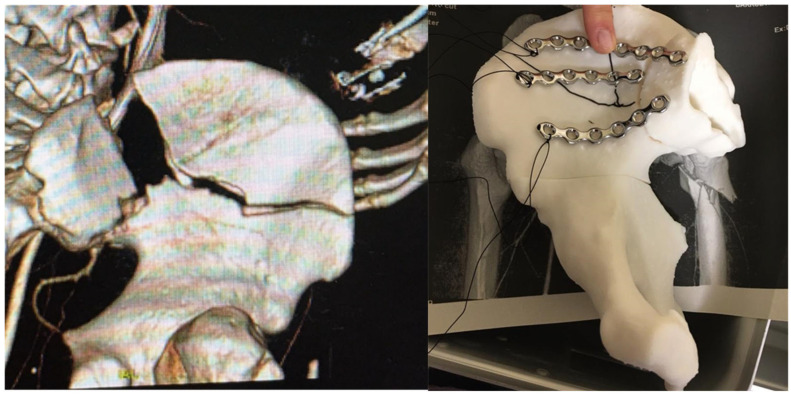
Anatomical model for pre-modeling synthesis plates for surgical treatment of acetabular fractures (traumatological surgery). **Left**: CT image, volumetric render of bone fracture. **Right**: Anatomical model for preformed osteosynthesis plates manufactured by SLA technology.

**Figure 6 healthcare-10-01791-f006:**
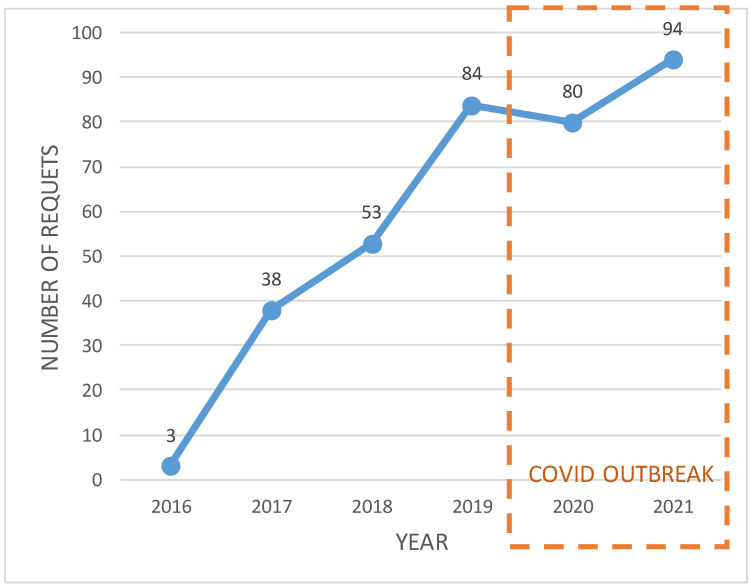
Evolution of requests registered on the intranet for 3D printing models at Cruces Hospital. It shows the requests received since the start-up of the service (September 2016) until the end of the present study (December 2021).

**Figure 7 healthcare-10-01791-f007:**
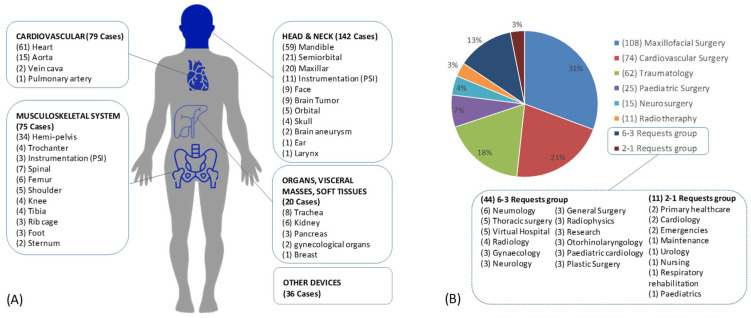
(**A**) Number of requests ranked by anatomy, grouped into 4 types of systems: head and neck, cardiovascular, musculoskeletal and soft tissues. Another group with other types of requests is also indicated. (**B**) Percentage and number of requests received by each hospital department. A total of 27 departments made at least one request. The departments with the fewest requests were grouped together: a group with 3 to 6 requests and a group with 1 or 2 requests. This figure was designed using resources from www.flaticon.com, accessed on 17 July 2022.

**Figure 8 healthcare-10-01791-f008:**
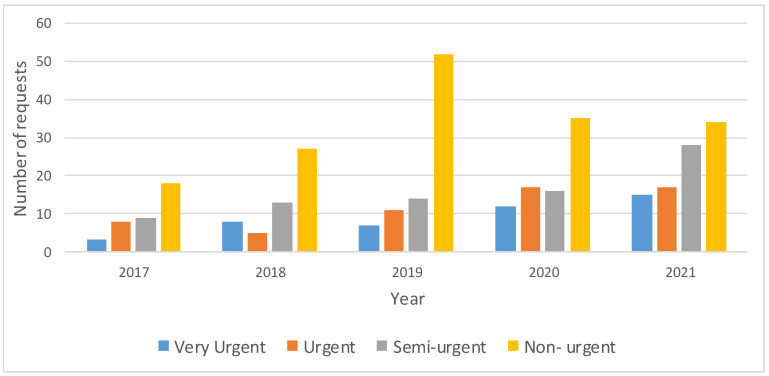
Evolution of the urgency of requests registered on the intranet for 3D printing models at Cruces Hospital. It shows the requests in complete years from 2017 to 2021. The 3 cases from 2016 were removed, as they are not considered representative.

**Figure 9 healthcare-10-01791-f009:**
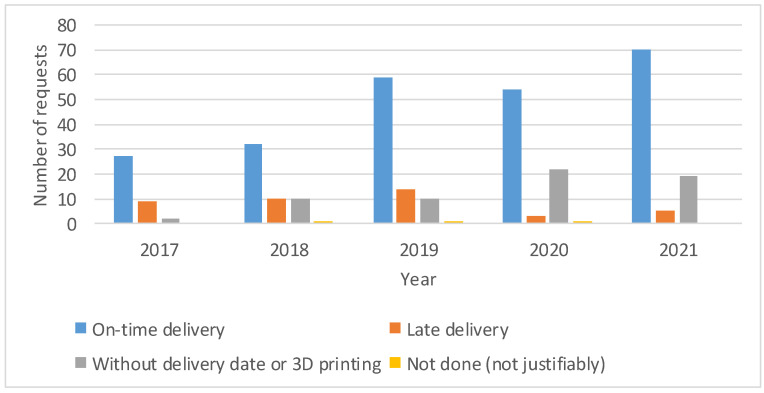
Evolution of the delivery success. Counts of cases in which the delivery date was earlier than the requested date (on-time delivery); in which the delivery date was later than requested (late delivery); in which no delivery date was recorded, or the 3D printing of the model was not considered necessary; and in which the model was not printed for a justified reason. The number of cases segmented by year is shown, as well as the percentage of the total number of cases in that year. The 3 cases from 2016 were removed as they are not considered representative for this figure.

**Table 1 healthcare-10-01791-t001:** Type of uses of requests for 3D models.

Type of Usage	Times	Times/Total Cases
Surgical Planning	190	54%
Surgical Training	125	36%
Premolding of osteosynthesis plates	82	23%
Guidance/support during surgery	91	26%
Teaching	85	24%
Patient communication	90	26%
Others	65	18%

**Table 2 healthcare-10-01791-t002:** Manufacturing errors.

Technology	Number of Cases	Number of Measures	Error Mean ± Sd
FDM	52	468	1.94 ± 0.05%
SLA	29	259	2.88 ± 0.06%
MJ	33	297	2.73 ± 0.03%

**Table 3 healthcare-10-01791-t003:** Material and machine costs.

Anatomy	Material Costs			Machine Time			Machine Cost (Amortization)			Total Cost		
	FDM	SLA	MJ	FDM	SLA	MJ	FDM	SLA	MJ	FDM	SLA	MJ
Mandible	EUR 0.70	EUR 11.60	EUR 58.10	4.6 h	5.5 h	6.1 h	EUR 1.10	EUR 2.37	EUR 88.88	EUR 1.80	EUR 13.97	EUR 146.98
Hemipelvis	EUR 5.70	EUR 51.90	EUR 379	40.3 h	19.8 h	18.7 h	EUR 9.67	EUR 8.51	EUR 272.46	EUR 15.37	EUR 60.41	EUR 651.46
Heart (1 chamber)	EUR 3.30	EUR 74.00	EUR 183	20.4 h	16.6 h	5.2 h	EUR 4.90	EUR 7.14	EUR 75.76	EUR 8.20	EUR 81.14	EUR 258.76

**Table 4 healthcare-10-01791-t004:** Technician time and costs.

Anatomy Model	Radiologist	Engineer	Manufacturing Technician	Human Resources Costs
Bone	30 min	30 min	50 min	EUR 70.96
Bone with mass	40 min	30 min	50 min	EUR 81.46
Visceral masses	180 min	30 min	50 min	EUR 228.41
Heart	150 min	40 min	50 min	EUR 202.92
Vascular anatomy	45 min	30 min	50 min	EUR 86.70
Trachea	45 min	30 min	50 min	EUR 86.70

**Table 5 healthcare-10-01791-t005:** Usability survey responses.

Question	Mean ± SD
*1. Have you used the 3D model for pre-planning the surgery? (Y = 25)*	
1.1. The printed model has provided me with relevant information that the digital model had not offered me.	4.36 ± 0.95
1.2. Surgical preparation using the model has had a positive effect on the final surgical outcome.	4.32 ± 0.95
1.3. The surgical approach has been modified after examination of the 3D model.	2.92 ± 1.32
1.4. The surgical instruments have been modified after examination of the 3D model.	2.28 ± 1.46
1.5. 3D models should be part of the planning for this pathology as a “gold standard” process.	4.04 ± 1.14
*2. Have you used the 3D model as a communication support with the patient? (Y = 13)*	
2.1. The patient/guardian has been more receptive/collaborative after explaining the pathology with their personalized 3D model.	4.15 ± 1.35
2.2. The patient/guardian has shown interest in keeping the 3D model.	2.08 ± 1.49
2.3. The 3D model has contributed to a better understanding of the intervention by the patient/guardian compared to other tools or drawings.	4.08 ± 1.31
2.4. The patient has been reassured to know that the surgery is being customized through a 3D model.	3.69 ± 1.50
*3. Have you used the model during the surgery? (Y = 15)*	
3.1. The 3D model has made it possible to avoid unforeseen events/complications during the operation.	3.93 ± 1.54
3.2. The 3D model has facilitated communication with colleagues.	4.73 ± 1.26
3.3. The 3D model has increased the confidence of the clinical team during the surgery.	4.33 ± 1.34
3.4. Having the 3D model in the operating theatre has proved useful.	4.60 ± 1.25
*4. After the surgery I consider that: (Y = 28)*	
4.1. The operation time was shorter than usual for this type of pathology	3.32 ± 1.47
4.2. The 3D model represents the anatomical area as observed during the operation.	4.39 ± 0.99
4.3. The same model on a smaller scale would have sufficient for the same purpose.	1.43 ± 0.88
4.4. The virtual model (3D PDF file) would have been sufficient for the same purpose.	1.68 ± 1.22

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
