# Peer review of "Implementation of an In-House 3D Manufacturing Unit in a Public Hospital’s Radiology Department"

_healthcare, 2022, doi:10.3390/healthcare10091791_

Round 1

Reviewer 1 Report

We thank the authors for the very interesting and particular work done. We think that the virtual reality and 3D printing will be an integral part in the future for the complex procedure in surgery  and Radiology.

Anyway we think there must be more information in terms of economic costs when we speak about new technologies in public health service. In line 86 and 93 you speak about cost-benefit and economic aspect but there are only a few or almost no data about the economic impact of this service in the daily clinical practice.

In table 5 (1.2) you write that surgical outcome have a positive effect using the 3D model but it is not clear  in terms of what. There are no data on hospitalization days, post-operative complications or shorter surgical procedure time.

In line 323 it is written that in 83,2% of the cases the request was for an anatomical model. Are the costs of this service justified for having only a 3D anatomical model when we actually can easily have 3D reconstructions on the TC images? Perhaps this new technology must be applied only for very selected cases or complex surgery procedure.

Furthermore  the numbering of paragraphs must be correct as well as the English editing and style.  

For these reasons we think that the article must be reconsidered after major revision.

Reviewer 2 Report

In the aspect of the new era of healthcare this subject is of high importance and articles summarizing clinical utility of different cases are valuable for further development of this area. In this article authors showed various cases in last six years that are performed in-house. Education, training, costs and other aspects of introducing in-house 3D printing in hospital is elaborated in details. There are valuable recently published articles that should be included in discussion (Micallef J et al, 2022; Beer N et al, 2022...). After minor modifications I recommend article for publication.
